# Patient-Perceived Quality Assessment in Orthopedics and Traumatology Departments during COVID-19 Pandemic

**DOI:** 10.3390/healthcare12090879

**Published:** 2024-04-24

**Authors:** Karoly Bancsik, Codrin Dan Nicolae Ilea, Mădălina Diana Daina, Raluca Bancsik, Corina Lacramioara Șuteu, Simona Daciana Bîrsan, Felicia Manole, Lucia Georgeta Daina

**Affiliations:** 1Faculty of Medicine and Pharmacy, Doctoral School of Biomedical Sciences, University of Oradea, 1 December Sq., 410081 Oradea, Romania; arcmaeoffice@yahoo.com; 2Faculty of Medicine and Pharmacy, University of Oradea, 1 December Sq., 410081 Oradea, Romania; 3Clinical Emergency Hospital “Avram Iancu”, 410027 Oradea, Romania; 4Department of Psycho-Neurosciences and Recovery, Faculty of Medicine and Pharmacy, University of Oradea, 1 December Sq., 410081 Oradea, Romania; 5Department of Surgical Disciplines, Faculty of Medicine and Pharmacy, University of Oradea, 410081 Oradea, Romania; fmanole@uoradea.ro

**Keywords:** COVID-19 pandemic, Patient-Perceived Quality Assessment (PPQA) questionnaire, quality of medical care, satisfaction, orthopedics and traumatology, coronavirus

## Abstract

The COVID-19 pandemic has led to significant disruptions in various healthcare systems. In Romania, the elective procedures in the orthopedic and traumatology specialty were one of the most affected. The study aims to investigate the influence of the COVID-19 pandemic on patient perceptions of quality in these departments. Standardized assessment tools were used, which consist of satisfaction questionnaires addressed to patients in order to assess the quality of health services in orthopedics and traumatology departments. Thus, a retrospective study was conducted using satisfaction questionnaires addressed to patients admitted to the orthopedics and traumatology departments of the County Clinical Emergency Hospital Bihor between January 2019 and December 2022. Eight reports, based on 746 questionnaires conducted during the studied period, were evaluated. To gauge patient satisfaction, Likert scales featuring five response options were used. A total of 627 questionnaires were valid, the exclusion criteria being incomplete questionnaires (the patients did not respond on all questions, n = 119). Four domains were analyzed: demographic data, hotel conditions, quality of medical care, and overall satisfaction. Demographic data highlight that patients exhibited an equitable distribution across residences, with 50.2% hailing from urban locales, while 53.5% (n = 333) were female. Regarding the overall impression, in 2020, there was a decline in the top rating of 5 compared to 2019, dropping to just 45.10% from 53.45%. Scores of 4 increased to 41.83%, while scores of 3 stayed under 8.5%. Scores of 2 and 1 were negligible. In 2021 and 2022, we can observe a sustained increase in the number of patients who awarded 5 points for overall impression and a decrease in the number of patients who awarded 4 points compared to previous years. The maximum difference between 2020 and 2021 and the period before and after this period was 27.24% (*p*-value < 0.001). The results indicate that while overall impressions of the hospital remained positive throughout the studied period, there were notable fluctuations in satisfaction levels during the pandemic. Patient satisfaction with attending physicians dipped in 2020 from 86.70% to 77.78% but recovered by 2022. The same trend can be observed with nurses and caregivers, as well as hotel services, during this period. These findings underscore the importance of addressing patient concerns and improving the quality of care delivery, particularly during times of crisis.

## 1. Introduction

The World Health Organization highlights the vital importance of delivering quality medical services within the healthcare system. It emphasizes that the activities of healthcare systems should be attuned to the requirements of the populace, ensuring that individuals are treated with dignity and respect [1]. Within hospital settings, meeting the needs of the population necessitates the evaluation of patient satisfaction, a facet governed by relevant legislation and the directives of the National Health Quality Management Authority (ANMCS) in Romania. At the national level, public hospitals have instituted mechanisms for gathering patient feedback, aimed at assessing satisfaction regarding service standards, adherence to patient rights, and the ethical comportment of medical and sanitary personnel. As an integral component of the ANMCS certification procedure, hospitals are mandated to administer patient satisfaction surveys [2,3].

The COVID-19 pandemic has caused significant disruptions across various sectors, including healthcare systems worldwide [4]. In particular, orthopedics and traumatology departments have faced unprecedented challenges as they strive to maintain quality patient care amidst the pandemic [5]. Patient satisfaction and perceived quality of care have emerged as critical indicators of healthcare system performance, especially during times of crisis [6].

As the COVID-19 pandemic continues to evolve, it is imperative to assess its impact on patient-perceived quality in orthopedics and traumatology departments. Understanding how the pandemic has influenced patient satisfaction and perceptions of care quality is essential for developing strategies to mitigate its effects and enhance future healthcare delivery [7].

This study aims to explore the impact of the COVID-19 pandemic on patient-perceived quality in orthopedics and traumatology departments, focusing on the County Clinical Emergency Hospital Bihor in Romania. By analyzing patient satisfaction questionnaires conducted during the studied period, we seek to identify trends and fluctuations in satisfaction levels and quality perceptions among patients receiving orthopedic and traumatology care.

## 2. Materials and Methods

### 2.1. Study Design

This study is part of complex doctoral research that aims to assess the quality of medical services provided in the departments of orthopedics and traumatology. In a prior publication, we established that the degree of patient overall satisfaction or general impression of the hospital is significantly influenced by the caliber of medical care provided by the doctors and the specific hotel conditions in the hospital premises. Moreover, the adoption of a uniform and standardized monitoring mechanism for performance evaluation within the orthopedics and traumatology departments, employing a 5-point Likert scale, enabled a precise evaluation of perceived quality [8]. 

To conduct the investigation, initial consent was sought and obtained pertaining to database access, following which data retrieval and processing were undertaken. Eight reports, derived from the evaluation of 746 questionnaires collected during the study duration, were scrutinized, with 627 deemed valid; incomplete questionnaires (n = 119) constituted the exclusion criteria. The reports were compiled based on two iterations of the satisfaction questionnaire: version A, employed during the period spanning 2019–2020, and version B, utilized from 2021 to 2022. The two versions of questionnaires are very similar and do not affect the study. The questionnaires were provided by County Clinical Emergency Hospital Bihor.

The two variants of questionnaires comprised sets of 37 and 40 standardized inquiries. They were devised in accordance with the monitoring mandates pertaining to patient satisfaction, as outlined in the framework agreement governing the provision of medical assistance within the Romanian healthcare system. These questionnaires encompass seven sections or domains, each containing 1–16 questions. The domains encompass demographic information, accessibility and admission procedures, facility conditions, quality of medical care, patient safety and rights, overall satisfaction levels, and lastly, opportunities for observations and suggestions. To gauge patient satisfaction, Likert scales featuring five response options were employed [9].

The present study offers a comparative analysis between the quality perceived by the patient in the hospital during the COVID-19 pandemic and the pre-pandemic period. The main goal is to identify if the pandemic affects the perceived quality in order to the develop sustainable quality management strategies in the future. To achieve this goal, we analyzed 2 domains: quality of medical care and overall satisfaction. 

### 2.2. Statistical Analysis

Chi-square test and *t*-test were utilized to assess disparities between groups concerning the general attributes of the study cohort. To test if there are statistically significant differences in patient satisfaction across the four years, a one-way ANOVA was used. To ascertain the correlation between patient’s general impression and quality of medical care or hotel conditions, linear regression analyses were employed. We used rating of 5 as outcome for the logistic regression. The results were considered significant at a *p*-value lower than 0.05. Data compilation and statistical analyses were conducted using Microsoft Word and Excel software applications, version Office Profesional Plus 2019 [10,11]. 

### 2.3. Participants

The investigation was conducted at the County Clinical Emergency Hospital Bihor (CCEHB) through the examination of patient satisfaction questionnaires. CCEHB, a tertiary-level public hospital situated in N-W Romania, serves as a healthcare provider for approximately 200,000 residents of the Municipality of Oradea and delivers emergency medical services to a territorial population of approximately 600,000 individuals.

The orthopedics and traumatology wards, characterized by a comparable bed capacity (33 beds in Orthopedics and Traumatology Ward 1 and 30 beds in Orthopedics and Traumatology Ward 2) and the provision of identical medical services were selected as the sample wards for the study. The average number of discharged patients from each department analyzed ranged between 1150 and 1300 annually.

## 3. Results

Of the 746 questionnaires applied during the analyzed period, 84% (n = 627) of the questionnaires were validated, 51.4% (n = 322) from Orthopedics and Traumatology Ward 1 (O1) and 48.6% (n = 305) from Orthopedics and Traumatology Ward 2 (O2). Patients exhibited an equitable distribution across residences, with 50.2% hailing from urban locales, while 53.5% (n = 333) were female. Among the 627 respondents, 79.4% (n = 498) furnished comprehensive details regarding their place of residence, whereas 95.9% (n = 601) provided complete information regarding their educational attainment. Within the subset of patients reporting on their education, 75.5% (n = 454) possessed a high school diploma or attained a higher educational degree, 16% (n = 96) had completed the 8th grade, and 8.5% (n = 51) had received primary education. A comprehensive overview of the sample characteristics is presented in Table 1.

In order to identify how the pandemic affects the perceived quality, we assessed the overall impression of the hospital during the period studied. We found that on average, 93.44% of patients had a good (3 points), medium-good (4 points) and very good (5 points) overall impression of the hospital during the period studied.

In 2020, there was a decline in the top score of 5 compared to 2019, dropping to just 45.10%. Scores of 4 increased to 41.83%, while scores of 3 stayed under 8.5%. Scores of 2 and 1 were negligible. The maximum difference between 2020/2021 and other years was 27.24% (ANOVA *p*-value < 0.001). In 2021 and 2022, we can observe a sustained increase in the number of patients who awarded 5 points for overall impression and a decrease in the number of patients who awarded 4 points compared to previous years, the data being presented in Figure 1. For both wards, there was a dip in the highest ratings (5 points) in 2020, likely due to the impact of the COVID-19 pandemic. However, subsequent years showed a recovery, with a notable increase in top ratings by 2022. The 4-point ratings fluctuated but generally indicated that the majority of patients were satisfied to a certain extent. Lower ratings (1–3 points) were consistently minimal, suggesting few negative impressions. This trend is visually depicted in Figure 2 and Figure 3. A percentage of 6.72% refused to answer this question.

During the analyzed period, an average of 96.89% of patients rated the quality of medical care provided by the attending physicians within the orthopedics and traumatology departments as good (3 points), medium good (4 points), and very good (5 points). Notably, during the timeframe corresponding to the COVID-19 pandemic (2020–2021), there was a discernible decrease in the proportion of patients awarding 5 points for the quality of care offered by attending physicians, coupled with a corresponding increase in the proportion of patients awarding 4 points, in comparison to the years 2019 and 2022. This trend is visually depicted in Figure 4. The maximum disparity between the 2020/2021 and the periods preceding or following was 12.65% (ANOVA test *p*-value < 0.001). For both wards, patient satisfaction had a slight decrease in 2020 but improved notably by 2022, with the highest percentage of patients giving 5-point ratings in that year. The trends indicate an overall high level of patient satisfaction with the attending physicians, with a particularly notable recovery in the year 2022. This trend is visually depicted in Figure 5 and Figure 6. Additionally, a percentage of 2.79% of patients declined to respond to this question. We applied a logistic regression model to assess the quality of medical care provided by the attending physician in relation with the patient’s general impression, and we found that there is a direct relationship between these two variables (*p*-value < 0.05).

On average, 95.57% of patients rated the quality of medical care provided by nurses as good (3 points), medium good (4 points), and very good (5 points) in orthopedics and traumatology departments during the studied period. In the period corresponding to the COVID-19 pandemic (2020–2021), we can observe a decrease in the number of patients who awarded 5 points for the quality offered by nurses and an increase in the number of patients who awarded 4 points compared to 2019 and 2022, the data being presented in Figure 7. The maximum difference between 2020/2021 and the period before or after was 12.48% (ANOVA test *p*-value < 0.001). Regarding the medical care provided by nurses in each department, we can observe a fluctuating satisfaction trend with nurse services over the years. O1 showed a decrease in the highest satisfaction in 2021 but recovered in 2022. O2 experienced a significant improvement in 2022 after a notable dip in 2020. The overall trend suggests that patient satisfaction with nursing care is high and improving, with very few patients expressing the lowest satisfaction. This trend is visually depicted in Figure 8 and Figure 9. A percentage of 3.93% refused to answer this question. We applied a logistic regression model to assess the quality of medical care provided by nurses in relation with the patient’s general impression, and we found that there is a direct relationship between these two variables (*p*-value < 0.05).

On average, 94.59% of patients rated the quality of medical care provided by caregivers as good (3 points), medium good (4 points), and very good (5 points) in orthopedics and traumatology departments during the studied period. In the period corresponding to the COVID-19 pandemic (2020–2021), we can observe a decrease in the number of patients who awarded 5 points for the quality offered by caregivers and an increase in the number of patients who awarded 4 points compared to 2019 and 2022, the data being presented in Figure 10. The maximum difference between 2020/2021 and the period before and after was 17.57% (ANOVA test *p*-value < 0.001). Regarding the medical care provided by caregivers on each department, we can observe fluctuating levels of patient satisfaction with caregivers across the years. O2 experienced an initial decrease in the highest satisfaction ratings in 2020 but a sharp recovery in 2022. O1 displayed a high level of satisfaction that peaked in 2020, slightly declined in 2021, and then increased again in 2022. Overall, satisfaction levels for caregivers show an upward trend by 2022 in both wards. This trend is visually depicted in Figure 11 and Figure 12. A percentage of 4.42% refused to answer this question. We applied a logistic regression model to assess the quality of medical care provided by caregivers in relation with the patient’s general impression, and we found that there is a weak relationship between these two variables (*p*-value = 0.16).

On average, 90.16% of patients rated the quality of hotel services in the hospital as good (3 points), medium good (4 points), and very good (5 points) in orthopedics and traumatology departments during the studied period. In the period of 2020–2021, we can observe a decrease in the number of patients who awarded 5 points for the quality hotel services in the hospital and an increase in the number of patients who awarded 4 points compared to 2019 and 2022, the data being presented in Figure 13. The maximum difference between 2020/2021 and the period before or after was 53.5% (ANOVA test *p*-value < 0.001). Regarding the quality of hotel services on each department, we can observe a fluctuation in patient satisfaction with hotel services across the studied years in both wards. In O1, there was a decline in the highest satisfaction ratings in 2021 followed by a significant recovery in 2022. O2 followed a similar trend, with a notable recovery in 2022. This trend is visually depicted in Figure 14 and Figure 15. A percentage of 8.2% refused to answer this question. We applied a logistic regression model to assess the quality of hotel conditions in the hospital in relation with the patient’s general impression, and we found that there is a weak relationship between these two variables (*p*-value = 0.62).

The assessment regarding the patient’s general impression and the quality of medical care or hotel conditions pointed that there is a strong relationship between the quality provided by the attending physician, nurses, and general impression. The results are presented in Table 2.

## 4. Discussion

The COVID-19 pandemic has significantly reshaped healthcare systems worldwide, leading to disruptions in various medical specialties, including orthopedics and traumatology. These departments, critical for diagnosing and treating musculoskeletal injuries and disorders, have faced unprecedented challenges during the pandemic. This discussion delves into the multifaceted impact of the COVID-19 pandemic on patient-perceived quality in orthopedics and traumatology departments, examining disruptions in services, safety concerns, telemedicine adoption, staffing issues, and opportunities for innovation.

Throughout the monitored period, a high percentage of patients (93.44%) reported favorable general impressions of the hospital quality, encompassing ratings from good to very good. In 2020, there was a noticeable drop in top scores to 45.10% and an increase in four-point ratings, with negligible low scores. 

In the domain of medical care, 96.89% of patients rated physicians positively on average, with a slight decline observed during the peak pandemic years. A logistic regression indicated a strong association between the quality of physician care and overall patient impression. Nurse care quality was similarly high, with an average of 95.57% positive ratings. A slight decline in the highest scores was noted during 2020–2021, and a recovery pattern was seen in 2022. For hotel services, 90.16% of patients provided positive ratings overall, despite a dip during 2020–2021. 

The COVID-19 pandemic necessitated a substantial reorganization of healthcare services, resulting in disruptions to routine care and the postponement of elective procedures in orthopedics and traumatology departments. Hospitals and healthcare facilities diverted resources and staff to address the influx of COVID-19 cases, leading to the cancellation or delay of non-urgent surgeries and appointments. In order to give priority to urgent care requirements and preserve healthcare resources, elective procedures, such as ligament repairs, joint replacements, and arthroscopic surgeries, were postponed [12,13,14].

The patients waiting for orthopedic operations were greatly affected by these disruptions. Patients who required imaging investigations, for example, to determine bone density, were also affected [15]. Delays in seeking therapy resulted in a reduction in quality of life, limited mobility, and prolonged pain for many patients. Furthermore, the postponement of elective procedures may have led to the progression of musculoskeletal conditions, necessitating more invasive interventions in the future. From the perspective of patient-perceived quality, the inability to access timely care can contribute to frustration, anxiety, and dissatisfaction with the healthcare experience [16,17,18,19].

Orthopedics and traumatology departments have close physical contact with patients and healthcare providers, creating concerns regarding COVID-19 transmission within hospital settings. Hospitals responded by implementing severe infection control procedures to safeguard the safety of patients and workers. These steps included the required use of personal protective equipment (PPE), the adherence to hand hygiene standards, the installation of COVID-19 screening processes, and increased sanitation practices. While these precautions were necessary to prevent the virus from spreading, they also created worries among patients about safety and infection risks. Patients may have been afraid to seek medical care or have orthopedic surgeries owing to concerns of catching COVID-19 in healthcare settings. Furthermore, the usage of PPE, particularly masks and face shields, may have hampered effective communication and interpersonal interactions between patients and healthcare personnel, thereby reducing patient satisfaction and perceived quality of care [20,21,22].

The COVID-19 pandemic has accelerated the implementation of telemedicine and virtual treatment in orthopedics and traumatology departments, giving an alternate method of delivering healthcare services while lowering in-person contact and the danger of viral transmission. Orthopedic healthcare practitioners utilized telemedicine systems to perform remote consultations, exams, and follow-up sessions for patients. While telemedicine provided a convenient and safe alternative to traditional in-person therapy, it also brought issues in terms of patient satisfaction. Some patients may have believed that telemedicine consultations lacked the personal touch and detailed review that face-to-face visits offered. Furthermore, the inability to perform physical examinations or diagnostic testing remotely may have restricted the breadth of telemedicine consultations, thereby influencing patient satisfaction and trust in the treatment received [23,24,25].

The COVID-19 pandemic put an unprecedented strain on healthcare systems, causing staffing shortages and increasing workloads in orthopedics and traumatology departments. Healthcare practitioners encountered enormous problems as they handled the responsibilities of caring for COVID-19 patients while still providing necessary orthopedic therapies. Staff shortages and increased demand for healthcare services lead to exhaustion, burnout, and moral anguish among orthopedic healthcare practitioners. Staffing shortages and workload demands may have had numerous consequences on patient perception of quality. Patients may have faced lengthier appointment wait times, delayed responses to requests, or rushed sessions with their healthcare professionals. Furthermore, healthcare personnel under substantial stress may have been less attentive or compassionate towards patients, thus influencing patient satisfaction and perceptions of treatment quality [26,27,28,29].

Despite the obstacles given by the COVID-19 pandemic, orthopedics and traumatology departments showed tenacity and adaptation in the face of the disaster. Healthcare providers have adopted creative techniques to improve patient care delivery and satisfaction. Virtual rehabilitation programs, remote monitoring equipment, and tele-rehabilitation services were created to help patients manage their musculoskeletal disorders from the safety and comfort of their own homes. These novel techniques not only ensured the continuity of treatment during the epidemic but also provided potential to improve patient-perceived quality in orthopedics and traumatology departments. Patients valued the ease and accessibility of virtual treatment alternatives, which removed the need for travel and reduced exposure to infectious agents. Furthermore, virtual care platforms gave patients more flexibility in arranging appointments and obtaining healthcare services, which contributed to increased patient happiness and engagement [30,31].

### Limitations of the Study

The current investigation is subject to several inherent limitations typical of survey-based research methodologies. Notably, while there were statistical variances observed in demographic data between the two wards under scrutiny, these discrepancies did not impact the outcomes of the study. Moreover, it is important to acknowledge that the data accessible for research purposes do not encompass the entirety of patients admitted and discharged during the period under investigation.

The aim of the study was not to make comparisons between questionnaire types but to assess the patient’s perception of the quality of the medical staff’s care, the hotel’s ser-vices, and the hospital’s overall impression. 

## 5. Conclusions 

The COVID-19 pandemic has had a profound impact on patient-perceived quality in orthopedics and traumatology departments. The analysis of patient-perceived quality in orthopedics and traumatology departments during the COVID-19 pandemic highlights fluctuations in satisfaction levels across various aspects of care. While overall impressions of the hospital remained positive, there were notable decreases in the quality ratings of medical care provided by attending physicians, nurses, and caregivers as well as hotel services during the pandemic period, and we also found that the level of patient overall satisfaction or general impression about the hospital is strongly dependent on the quality of medical care provided by the doctors and the nurses. These findings underscore the importance of addressing patient concerns and improving the quality-of-care delivery, particularly during times of crisis. Moving forward, it will be essential for healthcare providers to continue prioritizing patient safety, accessibility, and satisfaction to ensure high-quality care delivery in orthopedics and traumatology.

## Figures and Tables

**Figure 1 healthcare-12-00879-f001:**
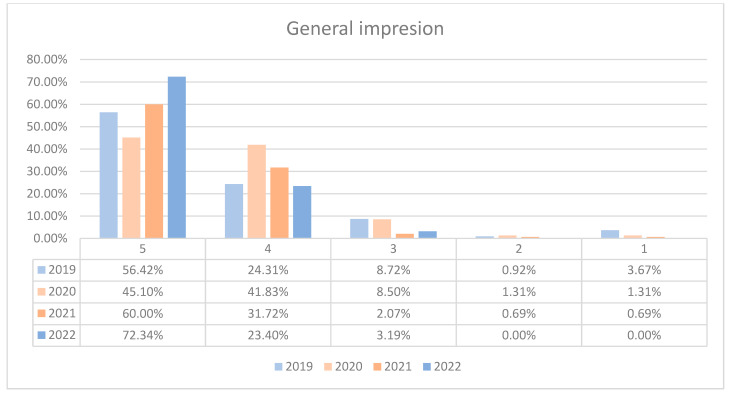
Responses regarding the general impression about the hospital reported annually.

**Figure 2 healthcare-12-00879-f002:**
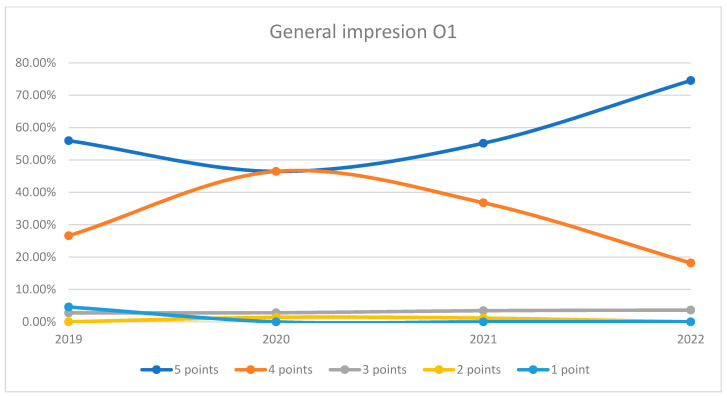
Responses regarding the general impression about the hospital in Orthopedics and Traumatology Ward 1 reported annually.

**Figure 3 healthcare-12-00879-f003:**
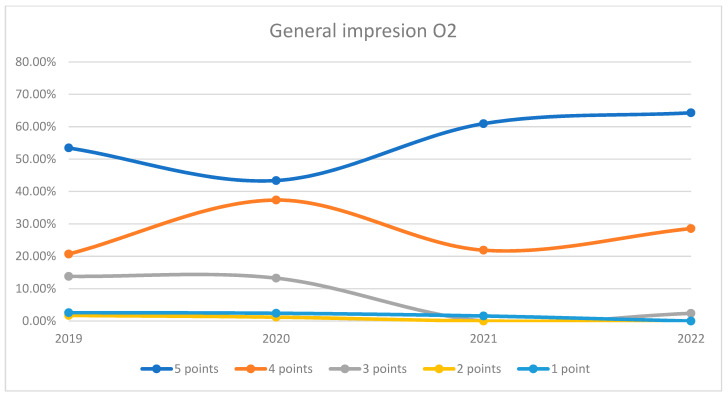
Responses regarding the general impression about the hospital in Orthopedics and Traumatology Ward 2 reported annually.

**Figure 4 healthcare-12-00879-f004:**
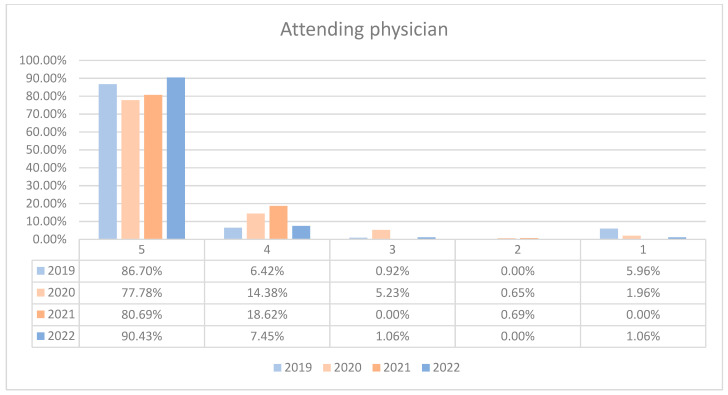
Responses regarding the quality of the medical care provided by the attending physician reported annually. 2019–2022.

**Figure 5 healthcare-12-00879-f005:**
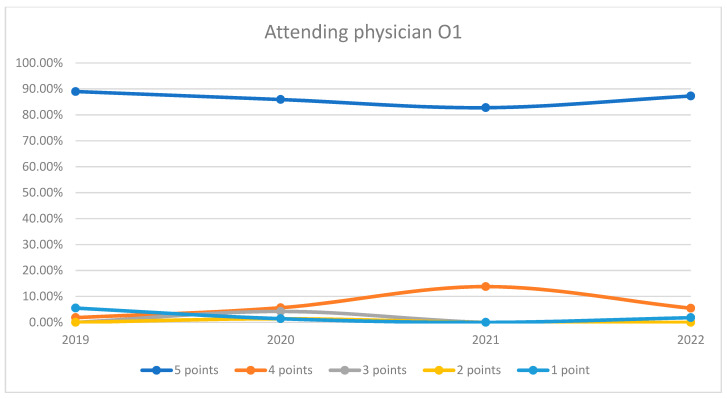
Responses regarding the quality of the medical care provided by the attending physician in Orthopedics and Traumatology Ward 1 reported annually. 2019–2022.

**Figure 6 healthcare-12-00879-f006:**
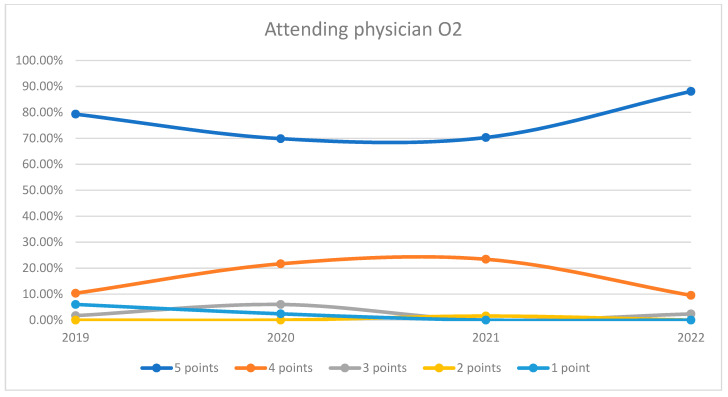
Responses regarding the quality of the medical care provided by the attending physician in Orthopedics and Traumatology Ward 2 reported annually. 2019–2022.

**Figure 7 healthcare-12-00879-f007:**
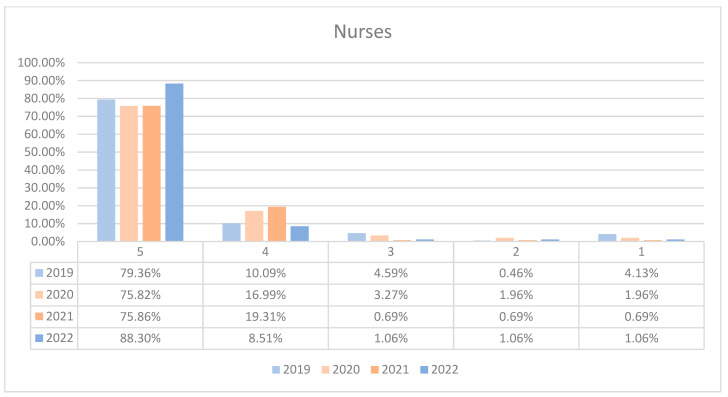
Responses regarding the quality of the medical care provided by nurses reported annually. 2019–2022.

**Figure 8 healthcare-12-00879-f008:**
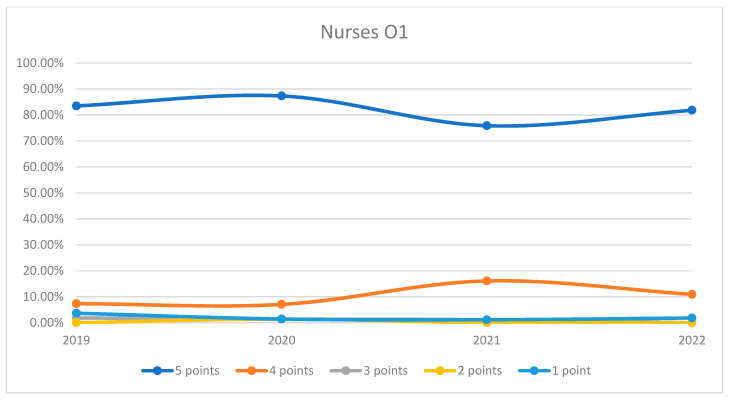
Responses regarding the quality of the medical care provided by nurses in Orthopedics and Traumatology Ward 1 reported annually. 2019–2022.

**Figure 9 healthcare-12-00879-f009:**
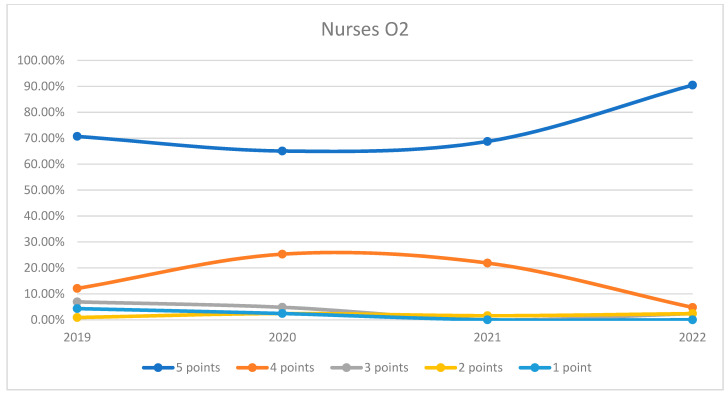
Responses regarding the quality of the medical care provided by nurses in Orthopedics and Traumatology Ward 2 reported annually. 2019–2022.

**Figure 10 healthcare-12-00879-f010:**
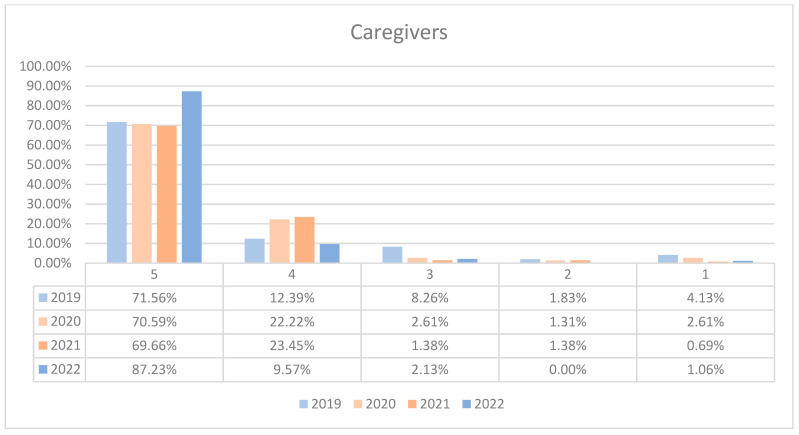
Responses regarding the quality of the medical care provided by the caregivers reported annually. 2019–2022.

**Figure 11 healthcare-12-00879-f011:**
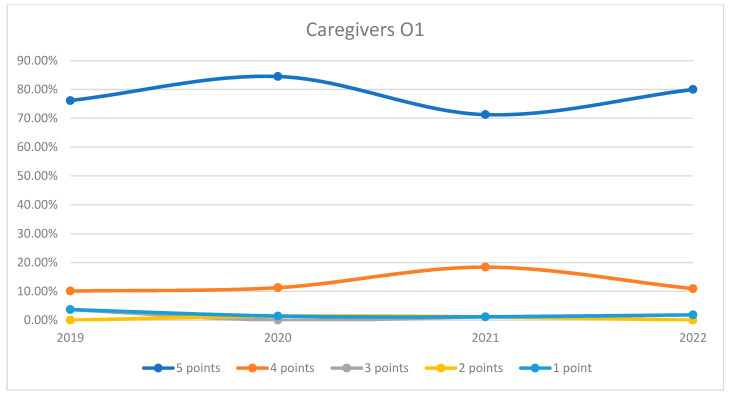
Responses regarding the quality of the medical care provided by caregivers in Orthopedics and Traumatology Ward 1 reported annually. 2019–2022.

**Figure 12 healthcare-12-00879-f012:**
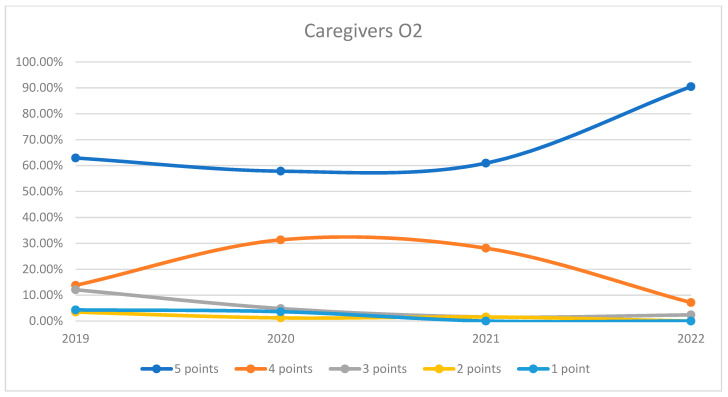
Responses regarding the quality of the medical care provided by caregivers in Orthopedics and Traumatology Ward 2 reported annually. 2019–2022.

**Figure 13 healthcare-12-00879-f013:**
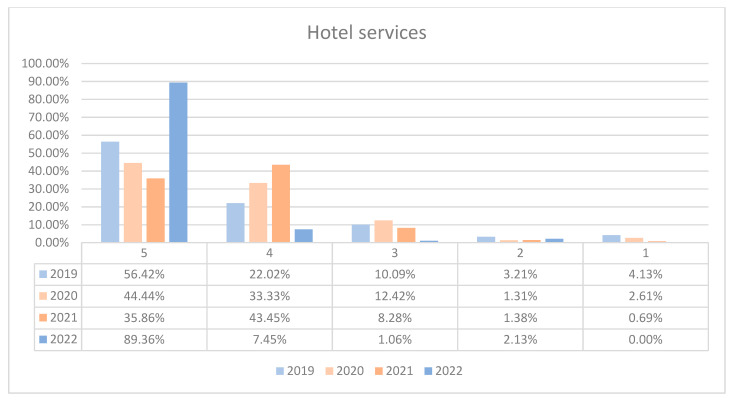
Responses regarding the quality of hotel services in the hospital reported annually. 2019–2022.

**Figure 14 healthcare-12-00879-f014:**
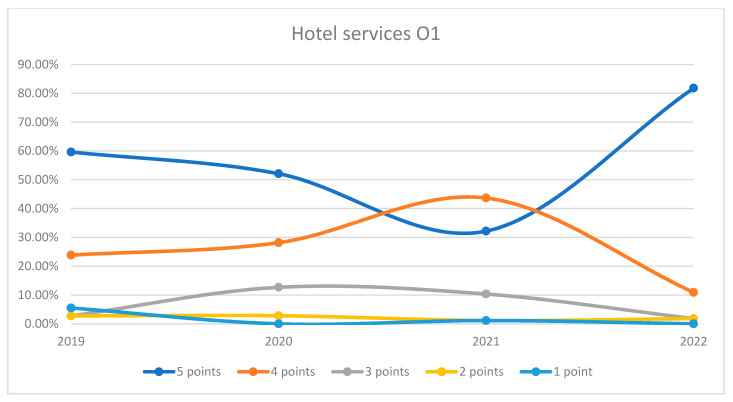
Responses regarding the quality of hotel services in Orthopedics and Traumatology Ward 1 reported annually. 2019–2022.

**Figure 15 healthcare-12-00879-f015:**
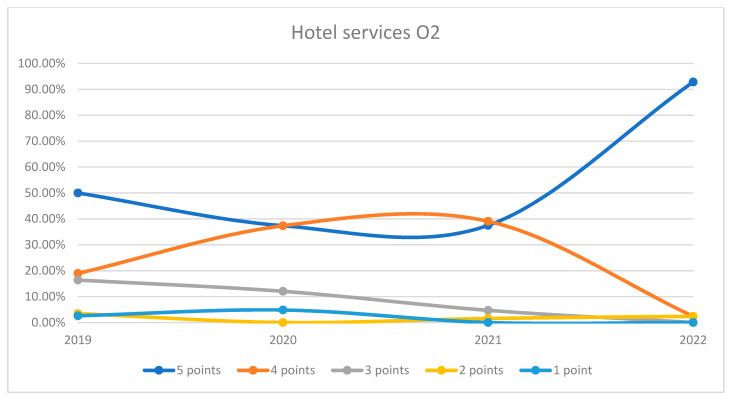
Responses regarding the quality of hotel services in Orthopedics and Traumatology Ward 2 reported annually. 2019–2022.

**Table 1 healthcare-12-00879-t001:** Demographic data.

Characteristic	N (%)			*p*-Value *
Department	Orthopedics 1 (O1)	Orthopedics 2 (O2)	Total Subjects	
Study sample	322 (51.4%)	305 (48.6%)	627	
Sex				0.52
Male	149 (23.9%)	141 (22.6%)	290 (46.5%)	
Female	171 (27.4%)	162 (26.0%)	333 (53.5%)	
Declined to answer	2 (0.3%)	2 (0.3%)	4 (0.6%)	
Residence				0.044
Urban	133 (26.7%)	117 (23.5%)	250 (50.2%)	
Rural	137 (27.5%)	111 (22.3%)	248 (49.8%)	
Declined to answer	52 (8.3%)	77 (12.3%)	129 (20.6%)	
Education				<0.001
Higher education	68 (11.3%)	59 (9.8%)	127 (21.1%)	
igh school diploma	184 (30.6%)	143 (23.8%)	327 (54.4%)	
8th class/grade	46 (7.7%)	50 (8.3%)	96 (16.0%)	
4th class/grade	16 (2.7%)	35 (5.8%)	51 (8.5%)	
Declined to answer	8 (1.3%)	18 (2.9%)	26 (4.1%)	

* Pearson’s chi-square test applied for each department.

**Table 2 healthcare-12-00879-t002:** Logistic regression results in relation with the patient’s general impression.

Category Correlation with General Impression	*p*-Value
Attending physician	<0.05
Nurses	<0.05
Caregivers	0.16
Hotel conditions	0.62

## Data Availability

Data are contained within the article.

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
