# Peer review of "Patient-Perceived Quality Assessment in Orthopedics and Traumatology Departments during COVID-19 Pandemic"

_healthcare, 2024, doi:10.3390/healthcare12090879_

Round 1
Reviewer 1 Report
Comments and Suggestions for Authors
Dear Respectable Authors
Thank you for considering a significant area of research related to the COVID-19 Pandemic and Patient-Perceived Quality in the Orthopedics and Traumatology Departments. Your results are of interest but your manuscript needs some revisions as follows;
- Please add a specific aim/purpose to the abstract section before the details of the methods.
- Please summarize lines 17-23 into three lines. Some statements are redundant.
- Abstract, please add more details regarding the methods you used including sample size, sampling methods, data collection period, variables, and the way you analyzed the data.
- Abstract, please add some demographic information (descriptive) at the start of the results and then state the analytic results.
- Abstract, please add the statistics related to each variable in front of it.
- Introduction, Lines 38-48, please separate the citations and place them in the right location. Please do not use group referencing.
- Lines 66-70, Please remove these statements here and replace them with the methods section.
- 2.1., some statements under this subheading are not related to study design and are relevant to setting and context. Please add a subheading for setting and replace related information )lines 73-82).
- Please add more details regarding sample size and sampling methods and how you measure it. It is not clear to me.
- Please remove Table 3 from the methods section and replace it with the results section.
- Please add a summary of the results at the beginning of the discussion section.
Cheers
Author Response
We really appreciate your recommendations. We tried, based on the recommendations, to improve and correct the article.
Attached you will find our answer for each point in the review.
Thank you.

Reviewer 2 Report
Comments and Suggestions for Authors
Upon reviewing the manuscript, several significant concerns regarding its scientific content have emerged. Firstly, the title fails to accurately encapsulate the comprehensive analysis conducted in the study. Moreover, the assertion of it being a complex doctoral project is not substantiated within the manuscript. Furthermore, the study suffers from inadequate justification and lacks theoretical underpinnings, rendering its methodological approach weak. The presentation of results is also lacking in clarity and depth. The study design is poorly elucidated, and the execution of the research appears convoluted. The utilization of two distinct questionnaires without clear rationale for comparison exacerbates confusion. Additionally, the sampling method is inadequately described, and the study criteria remain unspecified. Mention of data analysis focusing solely on univariate analysis overlooks potentially valuable insights. The exclusive focus on bar chart representations in describing findings neglects the richness of other data tables. Moreover, the rationale behind utilizing questionnaires to gauge the impact of the COVID-19 pandemic is questioned, suggesting the potential benefits of supplementing quantitative findings with qualitative insights. It is imperative to recognize the constraints imposed by pandemic-related healthcare shutdowns and governmental mandates, which may have impacted the study's feasibility and outcomes. Consequently, a comprehensive overhaul of the manuscript is warranted to enhance its scholarly value and contribute meaningfully to existing knowledge in the field.
Author Response

(The authors gave the same response as above.)

Reviewer 3 Report
Comments and Suggestions for Authors
11. Not sure the title is appropriate indicating “impact of the COVID-19” on quality by just assessing quality change over time prior and during the pandemic?
22. Table 1 should be considered a supplemental table. Table 1 generally describes the sample of respondents.
33. Only a single measure of overall patient satisfaction, many scales available.
44. Why two versions of the survey? What governed providing version A or B?
55. The author(s) could have modeled the longitudinal changes in patient satisfaction over time, adjusting for time-varying covariates.
66. The authors should indicate the statistical software used in the analyses. Microsoft Office is not a statistical package; I assume you used Excel?
77. Table 3 should really be table 1.
88. I would report the Chi-square value and df in table 3. Not sure I understand the footnote for table 3, the Chi-square test was conducted across the years not for each year?
99. Might have been interesting to assess satisfaction across time by department?
110. Covariates should have been included in the analyses, especially some indication of COVID influence?
111. While the authors indicate that 119 questionnaires were dropped due to missing data, and 627 had complete data? No missing values at all?
112. The authors indicate seven domain structures, are any of these domains composited (e.g., summed or averaged)? If this is so, how did you deal with bias due to questionnaire variants, especially since variant A was earlier times and variant B was later times for the quality of medical care domain?
T13. The authors may consider providing frequency distribution of the 1-5 scaling of perceived quality, instead of grouping scaling of 3, 4 and 5 as (93.44%).
114. Might have been interesting to have modeled a longitudinal ordered logit model on general impression.
115. The authors should define specific COVID time periods. For example, I assume 2019 data were from the pre-Covid period, was 2020 to 2022 considered the COVID period?
116. The authors indicate in lines 62- 63 page 2, “By analyzing patient satisfaction questionnaires from before, during and after the pandemic period…” are you assuming responses during 2022 were post-pandemic? Some of the highest daily cases reported in Romania occurred during 2022?
117. Since the authors assume that COVID-19 exposure may influence patient perceived quality and satisfaction, the assumption of constant COVID-19 exposure or perception of exposure to a patient may vary across time relative to reported cases and deaths. Did patients respond at different times during a year? If so, then assuming a constant COVID-19 exposure is probably not reasonable.
Author Response

(The authors gave the same response as above.)

Round 2
Reviewer 1 Report
Comments and Suggestions for Authors
Dear Respectable Authors
Thank you for your clarifications. There are several minor revisions as follows;
- Please Remove the supplementary file from the main text.
- Line 360, the reference number xxii is unclear.
- Reference section, you missed reference number 2.
- Also, the reference number vi must be rewritten as a number in the reference section.
Cheers
-
Author Response
We really appreciate your recommendations. We tried, based on the recommendations, to improve and correct the article.

Reviewer 2 Report
Comments and Suggestions for Authors
I looked over the new version of the paper. Thanks for considering all the feedback. You've made the paper better overall, and I'm happy with it.
Author Response
Thank you for your support and help. We look forward collaborating again with you.

Reviewer 3 Report
Comments and Suggestions for Authors
I think the authors should provide results from your regression model(s) in tabular form. Also, I think you could consolidate much of the graphics.

Author Response

(The authors gave the same response as above.)
